# Quantification of Age-Related Lung Tissue Mechanics under Mechanical Ventilation

**DOI:** 10.3390/medsci5040021

**Published:** 2017-09-29

**Authors:** JongWon Kim, Rebecca L. Heise, Angela M. Reynolds, Ramana M. Pidaparti

**Affiliations:** 1College of Engineering, University of Georgia, 597 DW Brooks Drive, Athens, GA 30602, USA; jongwonk@uga.edu; 2Department of Biomedical Engineering, Virginia Commonwealth University, Richmond, VA 23284, USA; rlheise@vcu.edu; 3The VCU Johnson Center, Virginia Commonwealth University Medical Center, Richmond, VA 23284, USA; areynolds2@vcu.edu; 4Department of Mathematics & Applied Mathematics, Virginia Commonwealth University, Richmond, VA 23284, USA

**Keywords:** aging, ventilator-induced lung injury, lung mechanics, pressure, computational models, tissue strains, tissue stress

## Abstract

Elderly patients with obstructive lung diseases often receive mechanical ventilation to support their breathing and restore respiratory function. However, mechanical ventilation is known to increase the severity of ventilator-induced lung injury (VILI) in the elderly. Therefore, it is important to investigate the effects of aging to better understand the lung tissue mechanics to estimate the severity of ventilator-induced lung injuries. Two age-related geometric models involving human bronchioles from generation G10 to G23 and alveolar sacs were developed. The first is for a 50-year-old (normal) and second is for an 80-year old (aged) model. Lung tissue mechanics of normal and aged models were investigated under mechanical ventilation through computational simulations. Results obtained indicated that lung tissue strains during inhalation (t = 0.2 s) decreased by about 40% in the alveolar sac (G23) and 27% in the bronchiole (G20), respectively, for the 80-year-old as compared to the 50-year-old. The respiratory mechanics parameters (work of breathing per unit volume and maximum tissue strain) over G20 and G23 for the 80-year-old decreased by about 64% (three-fold) and 80% (four-fold), respectively, during the mechanical ventilation breathing cycle. However, there was a significant increase (by about threefold) in lung compliance for the 80-year-old in comparison to the 50-year-old. These findings from the computational simulations demonstrated that lung mechanical characteristics are significantly compromised in aging tissues, and these effects were quantified in this study.

## 1. Introduction

Patients with obstructive lung diseases often receive mechanical ventilation to support their breathing and restore respiratory function. However, mechanical ventilation might lead to lung injury depending on the respiratory tract’s condition. The majority of patients requiring mechanical ventilation are elderly and advanced age is known to increase the severity of ventilator-induced lung injury (VILI) or ventilator-associated lung injury (VALI) [1]. The mortality rate for patients requiring mechanical ventilation is about 35% and this rate increases to about 53% for the elderly. The incidences of mechanical ventilation and in-hospital mortality both sharply rise with patient age [2]. Additionally, with increasing age, the severity of VILI increases [1]. Ventilator-associated lung injury is severer due to an increase in lung compliance [3]. Techniques for reducing VALI include the application of positive end expiratory pressure (PEEP), patient positioning, and low-volume and pressure-controlled ventilation [3]. Patients undergoing mechanical ventilation with inadequate fluid support are also at risk of developing hypovolemia which, over time, can lead to multiple organ system failure [4]. 

In general, with increasing age, the dynamic lung function and respiratory mechanics are compromised, and several experiments are being conducted to estimate these changes and understand the underlying mechanisms to better treat the elderly. Airway tissues are composed of heterogeneous material with mucosa and submucosa (epithelium), lamina propria or connective tissue, and smooth muscle layers that have varying properties [5,6,7]. These material properties in each layer of airway tissue vary with aging [8,9,10,11]. Several in vitro [12,13,14] and in vivo [15,16,17] models have been developed to study the effects of mechanical force or pressure on the airways. The changing dynamics of lung tissue in the elderly leads to an increase in lung compliance, and a decrease in susceptibility to atelectasis during mechanical ventilation [18,19,20,21]. Further, the elderly lung tissue exhibits increasingly dysregulated immune/inflammatory responses to injury leading to pathological increases in pro-inflammatory behavior [22,23].

Morphological tissue parameters of human lungs change due to aging, and these include, for example an approximate 30% reduction in the number of respiratory bronchioles between a 50-year-old and an 80-year-old [24]. Additionally, the alveolar sacs increase in size with age [25]. The initial shear modulus (stiffness) of tissue layer properties increased with age as reported in [8,9,10].

Several researchers previously measured the tissue strains responsible for stressing and rupturing the lung’s fiber network [26,27,28]. However, simulation studies of strains on lung tissue due to aging have not been extensively investigated. A review of the literature indicated that there are a few computational studies related to oxygen concentration in the blood [29,30] of lung airways but there are no computational studies of the strain environment in aging airways under mechanical ventilation. Therefore, it is important to investigate the aging tissue effects to better understand the severity of ventilator-induced lung injuries.

The purpose of this study is to investigate how the respiratory mechanics parameters of aging tissue are affected during mechanical ventilation. We hypothesized that aging greatly affects mechanical characteristics, and these effects are estimated through computational simulations for human bronchial and alveoli tissue models.

## 2. Materials and Methods

### 2.1. Tissue Models

The tissue model was assumed to be circular in shape representing the lower lung airway [31]. The airway tissue of bronchioles consists of three major layers (epithelium, lamina propria, and smooth muscle), whereas the alveolar sac consists of one single layer of epithelium. We considered the distinct composition of each layer for bronchiole with different material properties making up the tissue. The lumen diameter for the tissue and the thickness of tissue layers for the normal and elderly case were determined based on the literature review [32,33,34] for the computational tissue model in this study (Table 1). 

### 2.2. Tissue Dynamics

The governing equations for the tissue dynamics during mechanical ventilation are described by examining the forces acting on tissue, and are given as:(1)$$\rho \frac{Du}{Dt}=\nabla \cdot \sigma +f$$
(2)σ=Cε
where *ρ* is density, *u* is the displacement, *σ* is the Cauchy stress tensor, and *f* accounts for other external forces, *C* is the elasticity tensor and *ε* is the infinitesimal strain tensor (also called Cauchy strain tensor). 

### 2.3. Material Model

The lung tissue at the bronchioles and alveolar sacs is modeled as a hyperelastic material represented as a Neo-Hookean model involving a single parameter (initial shear modulus) based on measured test data [35]. In general, the Neo-Hookean material model is used for the tissue model in fluid-solid interaction (FSI) studies of the whole lung bifurcations [11]. However, for tissue analysis in this study, each of the layers of the tissue was assumed to be non-linear with hyperelastic material model in ANSYS (Canonsburg, PA, USA) software. In the simulation, we specified these material models at each layer with a different initial shear modulus. In a previous study, we used this approach to estimate tissue strains [36] and the results were reasonable. For the Neo-Hookean material model, the strain energy density function is given as:
*W* = *C*_1_ (*I*_1_ − 3)
where *C*_1_ is a material constant, and *I*_1_ is the first invariant of the right Cauchy-Green deformation tensor, i.e.:
*I*_1_ = *λ*_1_^2^ + *λ*_2_^2^ + *λ*_3_^2^
where *λ*_i_ are the principal stretches. *C*_1_ was used as the initial shear modulus in the hyperelastic Neo-Hookean model in ANSYS (Canonsburg, PA, USA). Initial shear modulus (material properties) values at each layer for the bronchiole region (G20) and alveolar region (G23) of the 50-year-old and the 80-year-old were calculated based on the literature review [33,36,37,38,39] and are presented in Table 2. 

### 2.4. Computational Simulations

The pressure waveform and material properties [33,36,37,38,39] were used in the tissue models simulation to obtain the tissue mechanics (strains and shear stresses). ANSYS (Canonsburg, PA, USA) Workbench was used to conduct the tissue structural simulation with ANSYS Mechanical (Version 15.0, Canonsburg, PA, USA). The lung tissue geometry was modeled using 6392 tetrahedral elements (6492 nodes) in the computational simulations. The input to the upper airway model (G1–G9) is the mechanical ventilation waveform, as shown in Figure 1a. The boundary condition (pressure) at the outlets for the upper airway was set to zero. However, for the lower airway model, the pressure obtained at the corresponding outlet was used as the boundary condition that was used for the tissue analysis. The lower lung airways model (G10–G23) was simulated by applying a mechanical ventilation waveform at the inlet (G10) which is obtained from the upper lung airways (G1–G9) model as shown in Figure 1a. Initial pressure and flow conditions applied to the lung airway models were the same for the 50-year-old and 80-year-old. More details regarding the inputs and boundary conditions for the upper and lower lung airway models can be found in Kim et al. [11]. Figure 1b shows the tissue geometry models for the 50 year-old and the 80 year-old at bifurcation G20 (bronchioles) and G23 (alveolar sacs). The single airway (terminal bronchiole) is connected to the entrance of the alveolar sacs. Each alveolus has an alveolar duct and numerous sacs. The results of pressure waveform obtained from lower lung airway simulations [11] were used in the tissue analysis models to obtain the tissue mechanical parameters (strains and shear stresses). 

The pressure waveform used for the heterogeneous tissue model at G20 and G23 is shown in Figure 2. The lower airway model was evaluated separately from the tissue model. The lower airways were modeled through FSI studies by treating tissue as one layer. Zero pressure boundary condition was used for the outlets in the simulation for the lower airway model. However, for tissue analysis, the pressure waveform obtained from the lower airway model at G20 and G23 (alveolar sacs) was used as the input to estimate the tissue strains. Conducting FSI studies of the lower airway model with tissue (having multiple layers) is challenging due to numerical instabilities in the simulations. Hence, we independently investigated the tissue analysis using the pressure waveform obtained from the lower airway model. Pressures obtained from lower bifurcation model were uniformly distributed and applied to tissue models to investigate the aging effects. At the truncated outlets (bifurcation G20 and others), a zero pressure boundary condition was assigned. This assumption is reasonable as it is not connected to others and is exposed. This is how the simulations were carried out so the airflow flows through airway bifurcations [11]. Airflow pattern predictions may have been influenced by the truncated airway outlets in the simulations. However, the pressure waveform obtained from the simulations for tissue analysis should be representative for estimating the tissue mechanical strains in this study. Computational simulations were carried out using ANSYS M echanical (Version 15.0, Canonsburg, PA, USA) for the structural analysis of tissue models to investigate the respiratory mechanics characteristics (stresses and strains). Figure 3 shows the overall simulation procedure adopted in this study.

## 3. Results

Results of lung mechanics (pressure and shear stress) and lung function (P–V loop) for normal and aged tissue models were obtained from the computational simulation and are discussed below.

### 3.1. Tissue Strains

Figure 4 and Figure 5 show the strains in each layer of the bronchiole (G20) and alveolar sac (G23) tissue for the 50-year-old and the 80-year-old at t = 0.2 s (mid-point of the inhalation) and t = 1.2 s (mid-point of the exhalation), respectively. The strain in Figure 4 and Figure 5 was obtained as change in deformation of the tissue model, and the strain percent (%) was computed as the maximum principal strain over the inhalation and exhalation period.

It can be seen from Figure 4a that the strains were highest (6.5%) at the epithelium, followed by the lamina propria (5.8%), and smooth muscle (4.2%) for the bronchiole tissue of the 50-year-old during inhalation. The strains for tissue layers for the 80-year-old were lower than that of the 50-year-old, 5.8% in the epithelium, 4.4% in the lamina propria, and 2.5% in the smooth muscle. Similarly, the strains in the tissue of the alveolar sacs for the 50-year-old were higher than the 80-year-old, as shown in Figure 4b. In general, the tissue strain for the 80-year-old significantly decreased by 27% at the bronchiole (G20), and by 40% in the alveolar sac (G23) at t = 0.2 s (mid-point of inhalation) due to lower tissue compliance in the 80-year-old in comparison to the 50-year-old. The pressures at which tissue strains are obtained, in general, are greater in the 80-year-old in comparison to 50-year-old. For example, pressure differences at the alveolar sacs and the bronchiole are 1.8 and 10 CmH_2_O. The strains during exhalation, as shown in Figure 5, were the highest in the epithelium, and the lowest in the smooth muscle. The trends for strain distribution during exhalation are similar to those observed in inhalation. 

Figure 6 and Figure 7 show the relationship between shear strain and shear stress in the bronchiole (G20) and the alveolar tissue (G23) during inhalation and exhalation, respectively. Shear strain can be defined as the area change of the deformation. Shear strain increased, corresponding to the increasing shear stress for bronchiole tissue during inhalation as shown in Figure 6a. A large shear stress difference of 33% (maximum) was observed between the 50-year-old and the 80-year-old. Shear strain at the alveolar tissue increased dramatically and the shear stress for the 50-year-old was 40% (maximum) higher than the 80-year-old during inhalation, as shown in Figure 6b. The shear strain at the bronchiole tissue also increased during exhalation, and the magnitude was lower than that obtained during inhalation. The maximum shear stress for the 50-year-old at bronchiole tissue was 50% higher than that of the 80-year-old as shown in Figure 7a. A similar trend was found in alveolar tissue as shown in Figure 7b.

Figure 8 and Figure 9 show the relationship between normal strain and normal stress in the bronchiole (G20) and the alveolar tissue (G23) during inhalation and exhalation. Normal strain was measured as the total displacement over initial length of the tissue. The stretch of the tissue was assumed to be normal or perpendicular to the model geometry. The normal strain increased until normal stress increased by approximately 2 Pa, and then the normal strain decreased even though normal stress was increasing at the bronchiole and alveolar tissue during inhalation, as shown in Figure 8a,b. Maximum normal strain for the 50-year-old was 15% higher than that of the 80-year-old at bronchiole tissue, and 30% higher at alveolar sac tissue as shown in Figure 8a,b. A similar trend was observed during exhalation as shown in Figure 9a,b. This result implies that normal strain was decreasing at some point and shear strain dominated in the tissue.

### 3.2. Sensitivity of Respiratory Mechanics Parameters 

Figure 10 shows the equivalent strain (calculated from the components of strain tensors) variation with age, indicating the sensitivity of tissue properties (thickness and material). For 50, 60, 70, and 80-year-olds, the geometrical parameters (diameter and centerline length) decreased by 10% in a decade, and the material properties increased by approximately 7% in a decade, respectively. These parameters were referred from a recent study by Kim et al. [11]. We used four data points (50-, 60-, 70-, and 80-years-old) for the regression equation, and it was interpolated using Akima interpolation method. Tissue thickness ratio is defined as the lamina propria maximum thickness divided by the epithelium minimum thickness. Similarly, the material properties change ratio is defined as the smooth muscle maximum stiffness divided by the epithelium minimum stiffness. The alpha (*α*) is defined as the ratio of material properties change ratio to tissue thickness change ratio. These parameters (equivalent strain and alpha) and their relationship changed with aging for both G20 and G23 as shown in Figure 10. This finding suggests that the tissue properties for the 80-year-old are more sensitive in comparison to the 50 year-old. 

Figure 11 shows respiratory mechanics parameters (work of breathing, compliance, and tissue strain) for the 50-year-old and the 80-year-old. The work of breathing is defined as pressure-volume (PV) loop area per unit tidal volume. As shown in Figure 11, the work of breathing decreased by about 64% over G20 and G23 during mechanical ventilation breathing cycle for the 80-year-old in comparison to the 50-year-old. Similarly, the maximum tissue strain decreased by about 80% over G20 and G23 for the 80-year-old. However, there was a significant increase (by about three-fold) in lung compliance for the 80-year-old in comparison to the 50-year-old. The results presented in Figure 11 illustrate that the airway mechanical parameters were impacted by aging.

## 4. Discussion

This study investigated the changes in lung mechanics parameters due to aging under mechanical ventilation for two geometric tissue models involving human bronchioles (G20) and alveolar sacs (G23). The two geometric tissue models corresponding to normal (50-year-old) and aged (80-year-old) cases were simulated computationally to estimate the relative changes related to aging. 

The strains in various tissue layers changed with age, and these changes were quantified during inhalation and exhalation as shown in Figure 4 and Figure 5. As shown in Figure 11, the strain decreased with age and this result is consistent with the trend shown in Figure 10. It is evident from the experimental data [40] that the strain in the lung tissue decreased with age. This observation was quantified through simulations, which indicated that there is a 40% decrease in strains on alveolar sac at t = 0.2 s for the 80-year-old as compared to the 50-year-old. Additionally, the shear stress decreased with age, specifically by about 40% from the 50-year-old to the 80-year-old and this result agreed well with the experimental data [41]. Tissue strains may cause cell injury [42,43], apoptosis, and necrosis [44]. The level of strain and the environment may determine whether these strains may lead to tissue injury. Our computational model may be helpful in determining the level of strains under a variety of mechanical ventilation conditions. 

The shear stress and shear strain relationship obtained from the present computational study at both the bronchiole and alveolar tissue is shown in Figure 6 and Figure 7 and represents an important finding as shear stresses/forces may be important for understanding the formation of fibrosis in tissues [45]. Shear stresses at tissue of lower bifurcation (G20) and alveolar sacs (G23) may enhance epithelial barrier function [46]. Therefore, it is essential to quantify the shear stresses to assess the effect of aging on the tissue. In the absence of experimental methods to estimate shear stresses during inhalation and exhalation, results from the present computational studies may provide an idea of stress levels in tissues. These levels of stresses in tissues may have an impact on cell behavior. 

Results of respiratory mechanics parameters obtained from the simulations indicated that the lower bifurcations were geometrically (diameter and centerline length) reduced by 10%, and the size of alveolar sacs was enlarged by 40% for the aging (80-year-old) model. As a result, there was a three-fold increase in the slope of PV curve from the 50-year-old to 80-year-old model in this study. Both the work of breathing and maximum tissue strain decreased by about 64% and 80%, respectively, for the 80-year-old in comparison to the 50-year-old. Interestingly, the lung compliance increased by threefold between the 50-year-old and 80-year-old. This suggests that the chest wall compliance which increases with age also impacted the work of breathing for the 80-year-old. We predicted a significant increase (about 25%) in the airway tissue stiffness with aging due to changes in material properties and geometrical variation, which is in agreement with previously-reported studies [11]. Some alterations in the microstructure of lung tissues with aging have also been previously reported, implying that changes in the charge density and size of proteoglycan and aggregation properties along with alterations in their distribution may be influencing the observed age-related changes in the biomechanics of tissues [47]. The age-related stiffening was also noticeable in the case of connective tissue (including lamina propria). Farzaneh et al. [10] reported the formation of fibrosis tissue with aging, which may play a part in the observed age-related stiffening of the connective tissue.

At lower tissue strain values for the 80-year-old, the strain in the bronchioles tissue was borne predominately by the elastic fibers in the connective tissue (lamina propria) [48]. Collagen in connective tissue, by contrast, was much stiffer and could only bear a small amount of strain before rupturing [49]. Overall, the simulation results indicate that for the 80-year-old there is about an 80% decrease in the strain of the tissue over G20 and G23 during the mechanical ventilation breathing cycle. This finding demonstrates the importance of including aging lung tissue strain information in evaluating the lung response to mechanical ventilation. This finding also suggests there is an increased sensitivity of the aged lung to VILI as the lung tissue strain directly reflects changes in lung mechanics and may be associated with VILI [2,50]. These strain levels may be susceptible to lung injury under mechanical ventilation for the elderly. The developed computational modeling studies reported here should be able to guide the experimental studies, as well as quantify the aging effects in human bronchioles and alveolar sacs tissues. These findings may also help develop age-dependent mechanical ventilation strategies to avoid VILI for the elderly in the future. 

## 5. Conclusions

This study investigated the effects of aging in tissue models in order to estimate the strain environments and respiratory mechanics parameters through computational simulations. For tissue analysis, a normal (50-year-old) and aged (80-year-old) tissue model were studied. Geometrical changes (tissue layer thickness) and material properties changes were considered as indicative of the aging effect, and the strain environment in the tissue was investigated. The simulation results showed that the strain distributions were substantially impacted by the change in model parameters. Overall, the results obtained from the simulations indicate that aging significantly affects lung function, and the strain environment may lead to tissue injury, which needs to be investigated further.

## Figures and Tables

**Figure 1 medsci-05-00021-f001:**
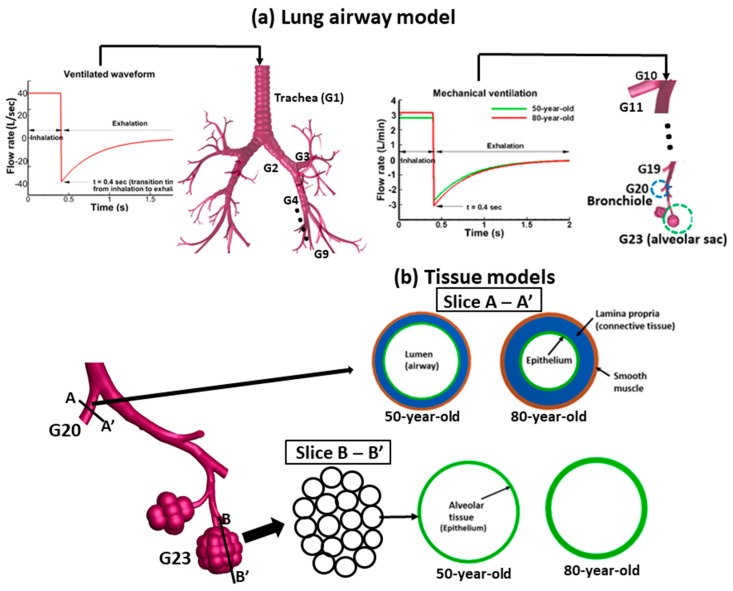
Overview of lung airway model and tissue model: (**a**) lung airways (G10–G23) and the applied waveform at the inlet; (**b**) tissue models of the bronchiole (G20) and the alveolar sac (G23). The lower bifurcation is the region where the aging effect has been considered. The single airway (terminal bronchiole) is connected to the entrance of the alveolar sacs. Each alveolus has an alveolar duct and numerous sacs.

**Figure 2 medsci-05-00021-f002:**
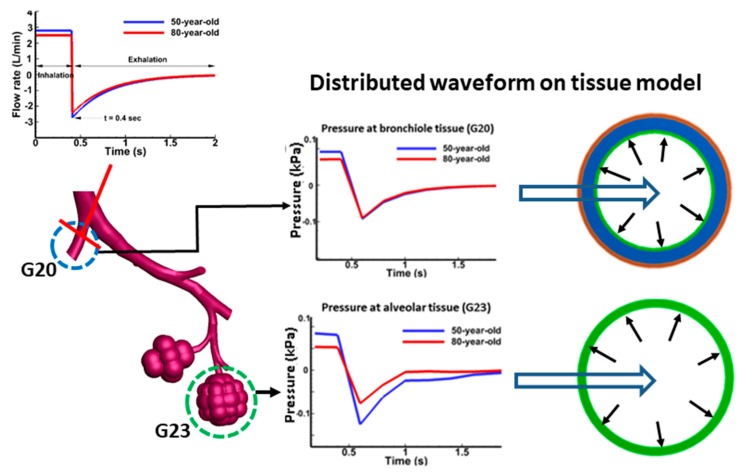
Pressure waveform for tissue model (G20 and G23) simulation. The pressure waveform was applied to tissue models for each age.

**Figure 3 medsci-05-00021-f003:**
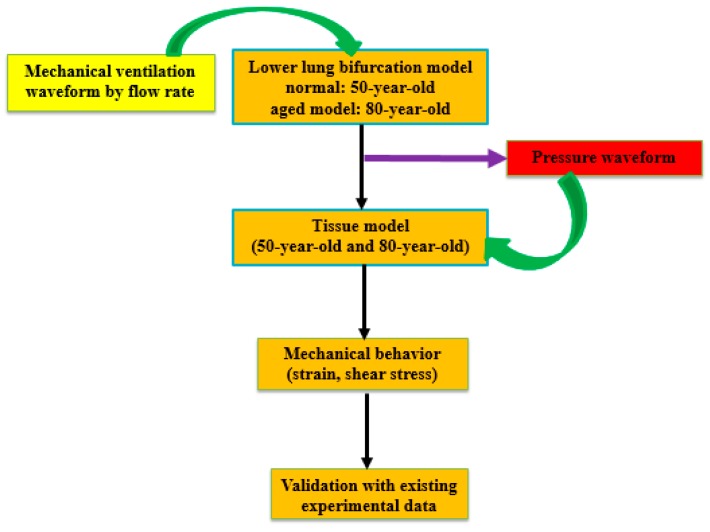
A schematic diagram of computational simulation of lung tissue with aging models.

**Figure 4 medsci-05-00021-f004:**
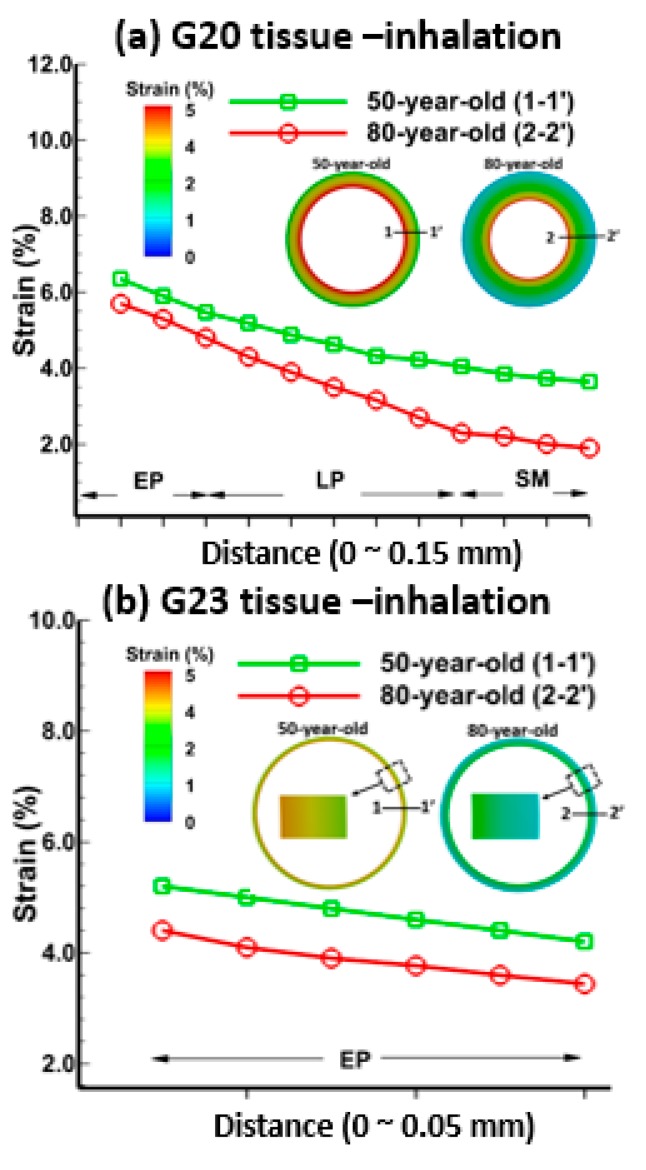
Tissue strain distributions during mid-point of inhalation (t = 0.2 s) for the 50-year-old and the 80-year-old at each layer—epithelium, lamina propria, and smooth muscle—of (**a**) the bronchiole (G20); and epithelium of (**b**) the alveolar sac (G23).

**Figure 5 medsci-05-00021-f005:**
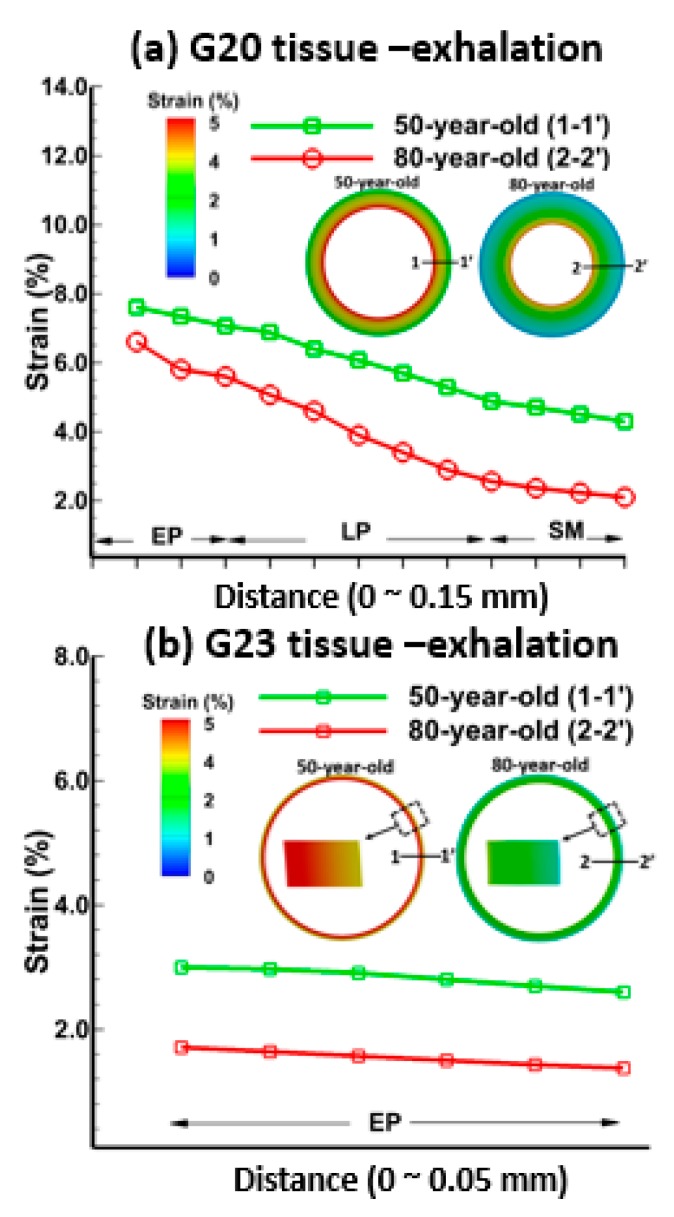
Tissue strain distributions during mid-point of exhalation (t = 1.2 s) for the 50-year-old and the 80-year-old at each layer—epithelium, lamina propria, and smooth muscle—of (**a**) the bronchiole (G20); and the epithelium of (**b**) the alveolar sac (G23).

**Figure 6 medsci-05-00021-f006:**
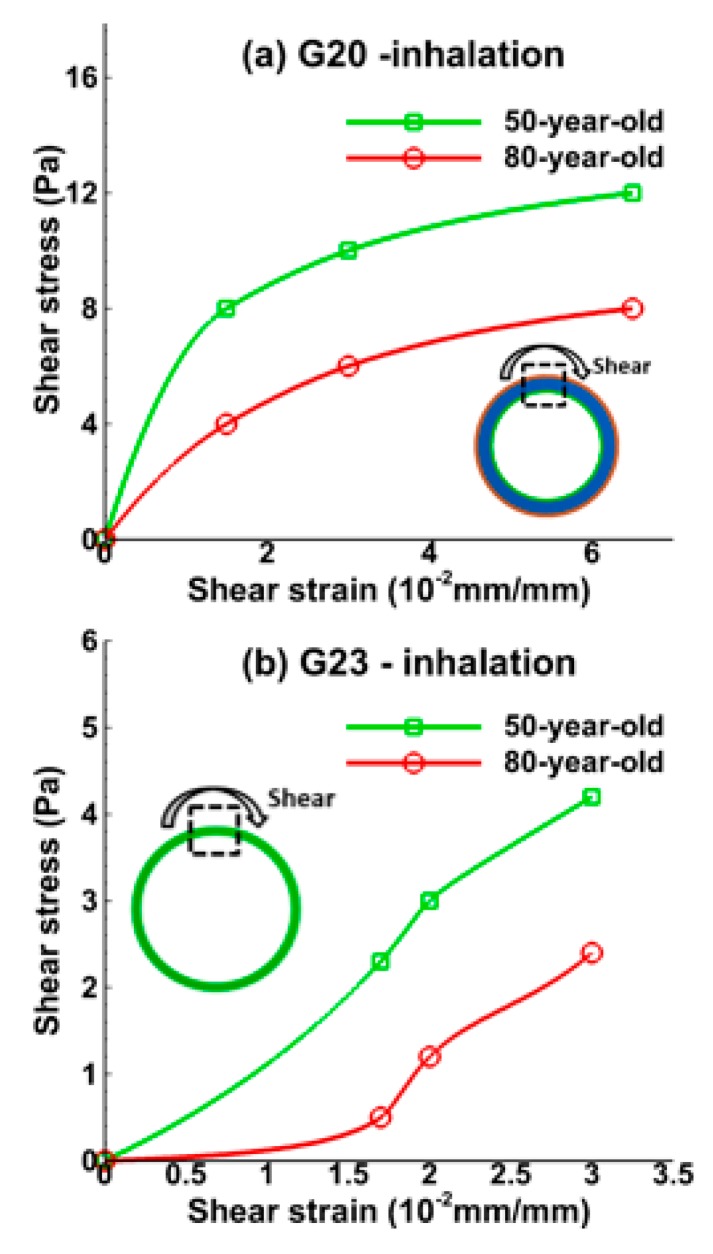
Relationship between shear strain and shear stress during mid-point of inhalation (t = 0.2 s) at (**a**) the bronchiole (G20) and (**b**) the alveolar sac (G23) tissue for the 50-year-old and 80-year-old.

**Figure 7 medsci-05-00021-f007:**
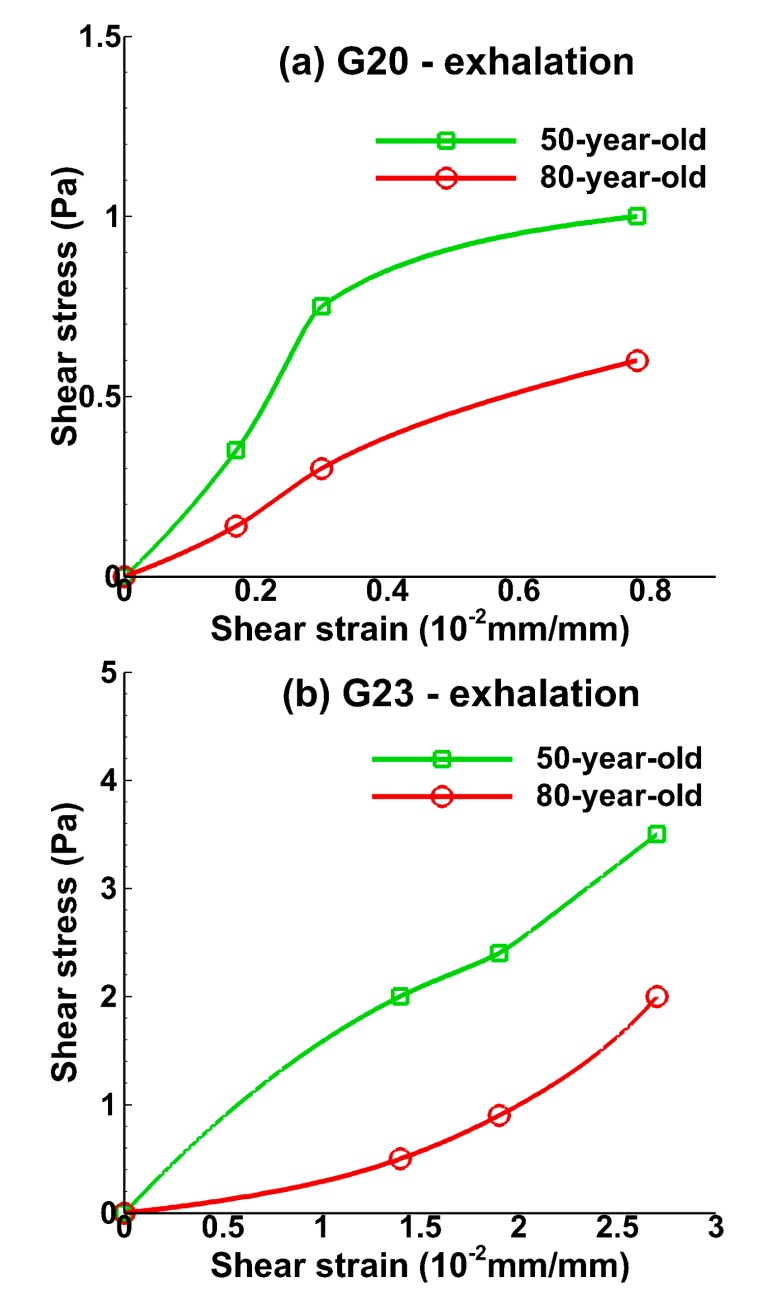
Relationship between shear strain and shear stress during mid-point of exhalation (t = 1.2 s) at (**a**) the bronchiole (G20) and (**b**) the alveolar sac (G23) tissue for the 50-year-old and 80-year-old.

**Figure 8 medsci-05-00021-f008:**
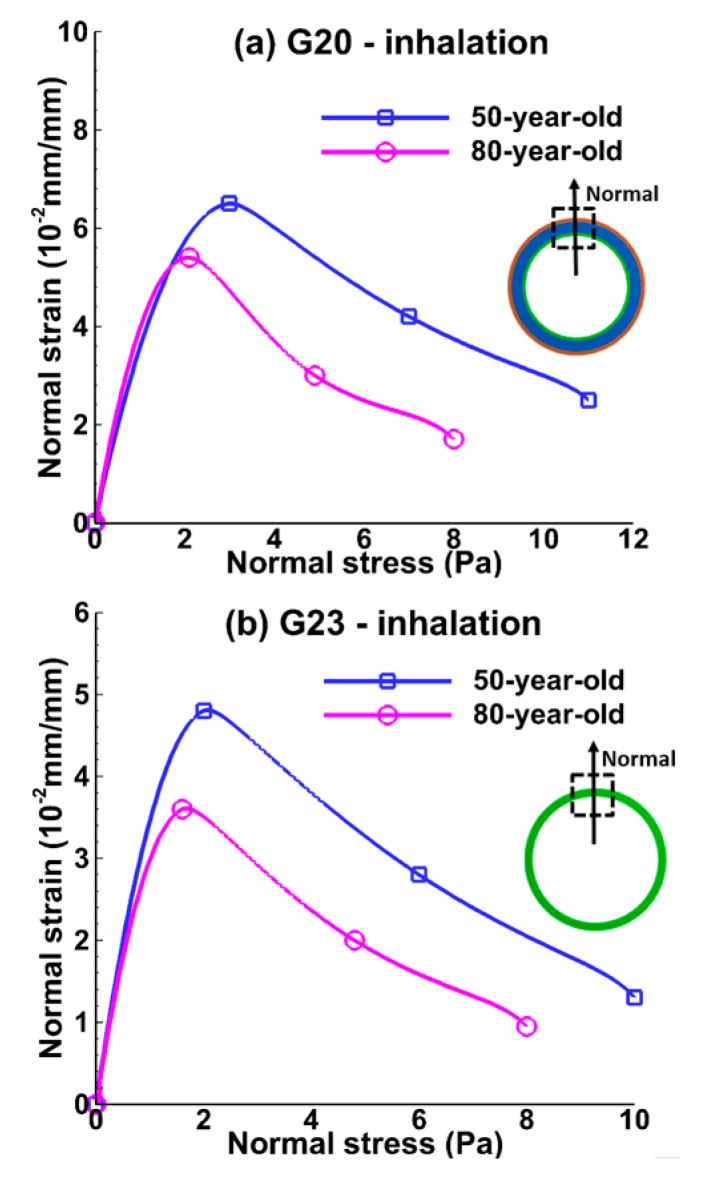
Relationship between normal strain and normal stress during the mid-point of inhalation (t = 0.2 s) at (**a**) the bronchiole (G20) and (**b**) the alveolar sac (G23) tissue for the 50-year-old and the 80-year-old.

**Figure 9 medsci-05-00021-f009:**
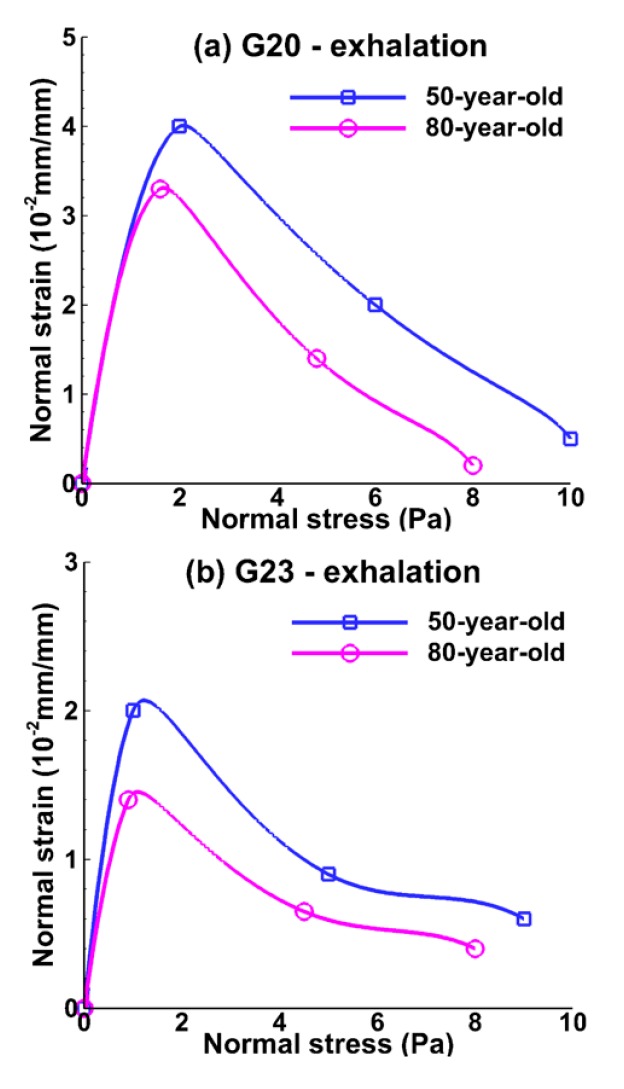
Relationship between normal strain and normal stress during mid-point of exhalation (t = 1.2 s) at (**a**) the bronchiole (G20) and (**b**) the alveolar sac (G23) tissue for the 50-year-old and the 80-year-old.

**Figure 10 medsci-05-00021-f010:**
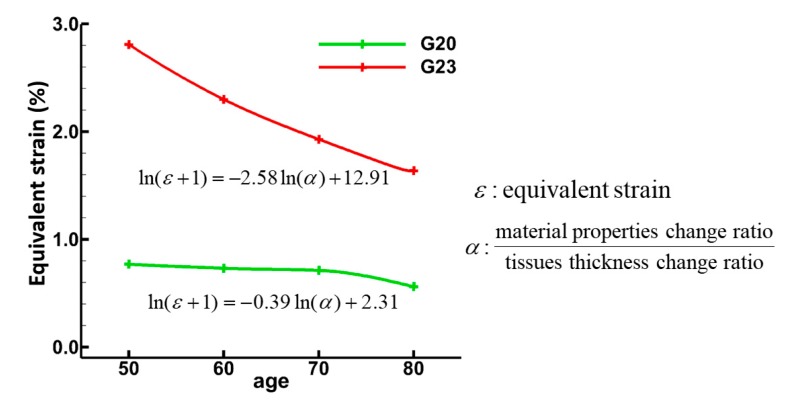
Material properties and tissue thickness sensitivity for normal and aged models. *ε* is equivalent strain and it allows any arbitrary three-dimensional strain state to be represented as a single positive value (scalar). Alpha (*α*) was defined as material properties change ratio to tissue thickness change ratio. Material properties change ratio was calculated as the smooth muscle maximum stiffness divided by epithelium minimum stiffness, and tissue thickness change ratio was calculated as the lamina propria maximum thickness divided by the epithelium minimum thickness.

**Figure 11 medsci-05-00021-f011:**
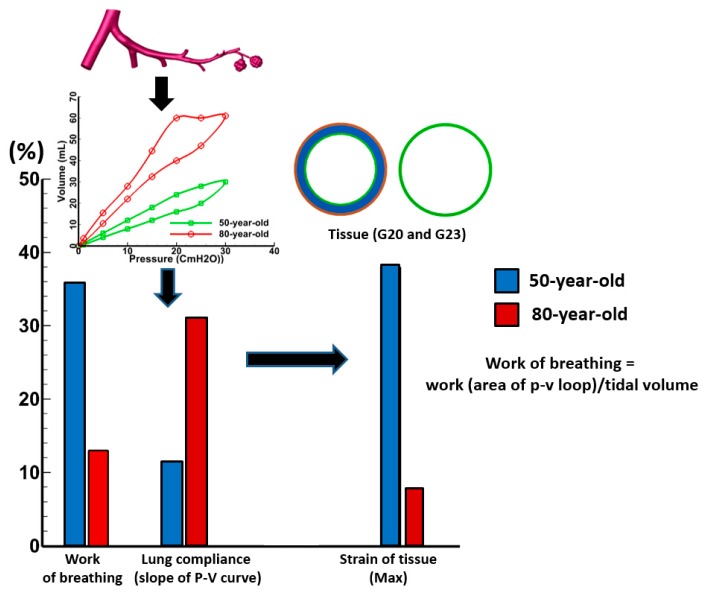
Comparison of respiratory mechanics parameters (work of breathing, compliance, and tissue strain) for the 50-year-old and the 80-year-old.

**Table 1 medsci-05-00021-t001:** Tissue morphological parameters at G20 and G23 for normal (50-year-old) and aged (80-year-old) models considered in this study.

**Parameters**	**Bronchiole Tissue (G20)**	
	**50-year-old**	**80-year-old**
Lumen diameter (mm)	0.45	0.32
Epithelium thickness (mm)	0.011	0.015
Lamina propria (mm)	0.056	0.073
Smooth muscle (mm)	0.016	0.021
	**Alveolar sac tissue (G23)**	
	**50-year-old**	**80-year-old**
Alveolus diameter (mm)	0.25	0.54
Alveolar tissue thickness (mm)	0.025	0.049

Note: The parameters were employed from the references [32,33,34].

**Table 2 medsci-05-00021-t002:** Material properties of each layer of tissue for 50-year-old and 80-year-old models.

		Initial Shear Modulus (Pa)		
		Epithelium	Lamina Propria	Smooth Muscle
50-year-old [33,36,37]	Bronchiole (G20)	112,000	145,600	160,000
	Alveolar sac (G23)	35,714	46,428	51,000
80-year-old [33,38,39]	Bronchiole (G20)	123,760	160,000	177,000
	Alveolar sac (G23)	39,464	51,300	56,400

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
