# Peer review of "Quantification of Age‐Related Lung Tissue Mechanics under Mechanical Ventilation"

_medsci, 2017, doi:10.3390/medsci5040021_

Round 1

Reviewer 1 Report

This is a well-organized paper focusing on the subject variability studies with different ages using CFD methods with some FSI techniques. A couple of suggestions are listed as follows and please make updates before this manuscript can be fully accepted:

(1) Equation numbers. the equations are without numberings,e.g. Line 93 to 96. Please correct this problem.

(2) For the slice B-B' shown in Fig. 1, it is unclear how a single airway can be connected with separated sacs. Please clarify this. 

(3) The truncated outlets (e.g., G20 in blue dash line and others) are shown in Fig. 2. However, the authors did not mention about the boundary conditions they use and did not justify that the truncated airway won't influence the accuracy of airflow pattern predictions.

I would like to review the revised version of this manuscript. 

Author Response

Reviewer #1 - Comments

Comments and Suggestions for Authors

This is a well-organized paper focusing on the subject variability studies with different ages using CFD methods with some FSI techniques. A couple of suggestions are listed as follows and please make updates before this manuscript can be fully accepted:

(1) Equation numbers. the equations are without numberings, e.g. Line 93 to 96. Please correct this problem.

Response: The equation numbers were added in lines 94 to 97 (page 3) of revised manuscript.

(2) For the slice B-B' shown in Fig. 1, it is unclear how a single airway can be connected with separated sacs. Please clarify this. 

Response: The single airway (terminal bronchiole) is connected to the entrance of alveolar sacs. Each alveolus has an alveolar duct and numerous sacs. This is added in Figure 1 caption (line 147 to 150 – page 4) of revised manuscript.

(3) The truncated outlets (e.g., G20 in blue dash line and others) are shown in Fig. 2. However, the authors did not mention about the boundary conditions they use and did not justify that the truncated airway won't influence the accuracy of airflow pattern predictions.

Response: At the truncated outlets (bifurcation G20 and others), a zero pressure boundary condition was assigned. This assumption is reasonable as it is not connected to others and is exposed. This is how the simulations were carried out so the airflow flows through airway bifurcations. Airflow pattern predictions may have been influenced by the truncated airway outlets in the simulations. However, the pressure waveform obtained from the simulations for tissue analysis should be representative for estimating the tissue mechanical strains in this study. This is added in lines 161 to 166 (page 5) in revised manuscript. 

Reviewer 2 Report

This is an interesting simulation demonstrating changes in the pulmonary mechanical stress environment as a result of aging. Changes in the stresses and strains caused by aging may have profound impacts on VILI. I have a couple of issues with the model formulation which may be addressed by clarifying the description. Of greater concern is the presentation and analysis of the results. For example, the authors seem to indicate that strain increases or decreases with age at different points in the manuscript. Calculations of work of breathing and tissue strain are provided at different lung volumes in the two groups and this does not seem like a fair comparison.

Introduction: Some of the references do not reflect the statements attributed to them. In particular, I was interested in how airway properties changed with aging (Line 52) so I consulted [9-11] and aging is not mentioned. Likewise, Line 63 (reference 9) does not discuss changes due to aging and the references in Line 64 [26-28] do not appear to address aging at all. This raises serious doubts about the rest of the references and I do not have the inclination to check them all.

Line 25: strains decreased with aging at what point in the respiratory cycle? What pressure?

Line 28: It’s hard to reconcile that the work of breathing increased with aging as the compliance decreased. It appears that the work of breathing was calculated at different tidal volumes which is not a fair comparison.

Lines 54-56: This is a confusing sentence since increased compliance, by definition, will result in increased distensibility.

Line 60: Would be more clear if this read ‘a 30% reduction in the number of lower bronchioles’. Also, lower bronchioles is not an anatomical term I am familiar with; perhaps the authors meant respiratory bronchioles?

Line 94: Please define rho, u, and C

Table 2: The increased stiffness with aging listed here seem to be in conflict with the statements in the introduction about reduced stiffness with aging. Please explain.

Line 103: Please justify the use of the same (neo-hookean) material model for all layers of the airway. It appears that this model was established for tissue as an a whole but the properties of each layer (e.g. epithelium vs smooth muscle) are quite different.

Line 124: What were the boundary conditions on the upper airway model (G1-G9)? Inspiratory pressure is easy to define based on the ventilation waveform but the outlet pressure seems like it would be dependent on the lower airway model.

Line 125: Was the same driving pressure/flow used for the 50 year old and 80 year old model?

Line 144: employed could be replaced with applied

Line 145: It appears that the lower airway model was evaluated in a simulation independent of the tissue (alveolar) model. Please provide boundary conditions for the lower airway model and justify why the lower airway model and tissue model were not coupled. It seems that the flow-induced pressure drop in the lower airways would be dependent on the alveolar tissue distensibility (e.g. infinitely stiff tissue would be 0 flow and 0 flow-induced pressure drop; infinitely compliant tissue would be high flow and high pressure drop).

Line 154: At what point in inspiration/expiration? Peak inspiration and end expiration?

Position in the respiratory cycle should be defined for all figures. Inspiration and Expiration are spans of time and the plots are of values at discrete time points.

The points in Figure 10 indicate that simulations were performed for ages 60 and 70. What were the model parameters for these runs? Epsilon must be defined.

Fig 10: ‘Material Properties Change Ratio’ and ‘Tissue Thickness Change Ration’ must be defined. Please justify the choice of regression equation; if there were just 2 data points (age 50 and 80) as indicated in the methods section why not just fit a straight line?

Fig 11: I have a number of concerns with this figure. 1) it should be made clear what direction the changes are. I can interpret from the PV loop that we are looking at the change from 50 to 80 year old models (e.g. compliance is greater in the 80 YO model). 2) it does not seem like a fair comparison to work of breathing at two different tidal volumes. There will be more work at a greater tidal volume even in the same lung model. In order to rectify this the work of breathing should be provided as work per unit volume (as is the standard practice). Also, the effort to overcome elastic recoil (not just resistive forces or PV loop area) should be considered since the effort of inspiration is not recovered during passive expiration. The formula used for work of breathing and the calculated values should be provided. 3) there were obviously be more strain in the 80 YO model because the lung volume is twice as high, I would prefer to compare tissue strains at the same lung volumes. 4) which tissue strain (G20 or G23) is shown?

Line 245: The discussion of strain is confusing since we don’t know what lung volume we are discussing. Also, Fig 10 shows strain decreasing with age and Fig 11 shows strain increasing with age.

Line 246: Does shear stress increase or decrease with age?

Line 252: I do not see a reference [66] in the bibliography

Line 255-257: Please discuss the ‘insights into tissue strains and the impact on tissue/cell function’ that your model may provide. The / is not proper grammar.

Line 259: I don’t understand what ‘lower bifurcations were reduced by 10%, and enlarged by 40% in the alveolar sac for the aged’ means.

Line 260: 23% change in what? volume delivered at a given P? Change in what direction?

Line 261: does ‘lower bifurcations’ mean ‘distal airways’

Line 263: Unless it’s shown that tidal volumes increase in aging I disagree with the statement about the work of breathing. Of course it takes more work to take a larger breath. The influence of the chest wall (which increases in stiffness with aging) on the work of breathing should also be mentioned.

Line 265: 40% change in what direction (increase or decrease with age)? At what lung volume?

Line 265: observed should be predicted

Line 266: Tissue stiffness seems like a prescribed parameter. Do the authors mean airway/alveolar stiffness (the stiffness of all types of tissue taken together)?

Line 266: Reference 46 seems to be in conflict with the statements in the introduction about tissue stiffness decreasing with age.

Line 278: What mechanical charecteristics of the alveolar tissue?

Line 280: Since there were only two model runs how is it possible to have a 10-40% change in compliance? Fig. 11 seems to indicate that the compliance change is a single scalar value.

Line 281: I don’t understand what ‘This finding indicates the effect of strain change’ means

Line 293: I would argue that the strain distributions were substantlly impacted by the change in model parameters and not ‘the aging model was significantly impacted by the strain distributions’.

Line 294: the conclusion that ‘These strains can greatly damage the tissue leading to shearing and swelling’ is not supported by the data. First, it is not shown that strain leads to swelling. Second, it seems that shearing would be the deformation of the tissue by shear stress which the authors argue earlier will lead to improved barrier function. Finally, does the model show increased or decreased strain in the 80 year old?  

Line 297: Fig 10 shows decreased strain and Fig 11 seems to show increased strain with age. Which is it? Assuming that strain goes down, please discuss this finding in context of the increase sensitivity of the aged lung to VILI. 

Author Response

Reviewer #2 - Comments

Comments and Suggestions for Authors

This is an interesting simulation demonstrating changes in the pulmonary mechanical stress environment as a result of aging. Changes in the stresses and strains caused by aging may have profound impacts on VILI. I have a couple of issues with the model formulation which may be addressed by clarifying the description. Of greater concern is the presentation and analysis of the results. For example, the authors seem to indicate that strain increases or decreases with age at different points in the manuscript. Calculations of work of breathing and tissue strain are provided at different lung volumes in the two groups and this does not seem like a fair comparison.

Introduction: Some of the references do not reflect the statements attributed to them. In particular, I was interested in how airway properties changed with aging (Line 52) so I consulted [9-11] and aging is not mentioned. Likewise, Line 63 (reference 9) does not discuss changes due to aging and the references in Line 64 [26-28] do not appear to address aging at all. This raises serious doubts about the rest of the references and I do not have the inclination to check them all.

Response: We thoroughly checked all the references again, and made sure the references are consistent with the statements in the revised manuscript. We mistakenly used the wrong references from a previous paper. Please see revised Refs. [9-12] in the revised manuscript.

Line 25: strains decreased with aging at what point in the respiratory cycle? What pressure?

Response: Strains decreased with aging at t = 0.2 s (inhalation), and at a pressure difference of 10 CmH2O between 50-year-old and 80-year-old for the bronchiole, and 1.8 CmH2O for the alveolar sacs. This is added in lines 25 to 28 (page 1) of the revised manuscript.

Line 28: It’s hard to reconcile that the work of breathing increased with aging as the compliance decreased. It appears that the work of breathing was calculated at different tidal volumes which is not a fair comparison.

Response: The work of breathing defined as area of the P-V loop per unit tidal volume is presented in revised Fig. 11 for fair comparison as suggested. The new results indicated that work of breathing for the 80-year old is reduced by 64% as compared with the 50-year old. This is stated in the revised manuscript. Please see revised Fig. 11., and lines 28 to 30 (page 1) of the revised manuscript.

Lines 54-56: This is a confusing sentence since increased compliance, by definition, will result in increased distensibility.

Response: Lines 54 to 56 is now restated in lines 57 to 59 (page 2) of the revised manuscript.

Line 60: Would be more clear if this read ‘a 30% reduction in the number of lower bronchioles’. Also, lower bronchioles is not an anatomical term I am familiar with; perhaps the authors meant respiratory bronchioles?

Response: Line 60 is now clearly restated in lines 63 to 64 (page 2) of the revised manuscript. We agree with the reviewer’s comment, and the term ‘lower’ is replaced with ‘respiratory’ in the revised manuscript.

Line 94: Please define rho, u, and C

Response: rho, u and C are defined in lines 98 to 100 (page 3) of the revised manuscript.

Table 2: The increased stiffness with aging listed here seem to be in conflict with the statements in the introduction about reduced stiffness with aging. Please explain.

Response: The conflicting statement in the introduction is restated with correct references to show the aging effects on stiffness/properties. Please see lines 64 to 65 (page 2) of the revised manuscript. This is now consistent with properties in Table 2.

Line 103: Please justify the use of the same (neo-hookean) material model for all layers of the airway. It appears that this model was established for tissue as an a whole but the properties of each layer (e.g. epithelium vs smooth muscle) are quite different.

Response: The authors agree with the reviewer that Neo-Hookean model is used for the tissue model in FSI (Fluid-Solid Interaction) studies of the whole lung bifurcations. However, for tissue analysis, each of the layers of the tissue was assumed to be non-linear with hyperelastic material model in ANSYS software. In the simulation, we have specified these material models at each layer with different initial shear modulus. In a previous study, we have used this approach to estimate tissue strains [37] and the results are reasonable. This is reflected in the revised manuscript. Please see lines 105 ~ 110 (page 3) of the revised manuscript.

Line 124: What were the boundary conditions on the upper airway model (G1-G9)? Inspiratory pressure is easy to define based on the ventilation waveform but the outlet pressure seems like it would be dependent on the lower airway model.

Response: The input to the upper airway model (G1-G9) is the mechanical ventilation waveform. The boundary condition of outlets for upper airway was set to zero. However, for the lower airway model, the pressure obtained at the corresponding outlet was used as the boundary condition that was used for the tissue analysis.  This is reflected in the revised manuscript. Please see lines 130 ~ 133 (page 4) of the revised manuscript.

Line 125: Was the same driving pressure/flow used for the 50 year old and 80 year old model?

Response: Yes, the same driving flow was used for the 50 year old and 80 year old models. This is added in line 136 to 137 (page 4) of the revised manuscript.

Line 144: employed could be replaced with applied

Response: the term ‘employed’ is replaced with ‘applied’ for figure 2 caption in line 172 (page 5) of the revised manuscript.

Line 145: It appears that the lower airway model was evaluated in a simulation independent of the tissue (alveolar) model. Please provide boundary conditions for the lower airway model and justify why the lower airway model and tissue model were not coupled. It seems that the flow-induced pressure drop in the lower airways would be dependent on the alveolar tissue distensibility (e.g. infinitely stiff tissue would be 0 flow and 0 flow-induced pressure drop; infinitely compliant tissue would be high flow and high pressure drop).

Response: Yes, the lower airway model was evaluated separately from the tissue model. The lower airways were modeled through FSI (fluid-solid interaction) studies by treating tissue as one layer. Zero pressure boundary condition was used for the outlets in the simulation for lower airway model. However, for tissue analysis, the pressure waveform obtained from lower airway model at G20 and G23 (alveolar sacs) was used as the input to estimate the tissue strains. Conducting FSI studies of lower airway model with tissue (having multiple layers) is challenging due to numerical instabilities in the simulations. That is the reason, we independently investigated the tissue analysis using the pressure waveform obtained from the lower airway model. This is added in line 151 to 158 (page 4 and 5) of revised manuscript.

Line 154: At what point in inspiration/expiration? Peak inspiration and end expiration?

Response: We have a breathing cycle for 2 seconds, 0 ~ 0.4 s is an inhaled time and 0.4 ~ 2 s is an exhaled time. This is an exact middle point of inhalation and exhalation. This is added in lines 182 to 183 (page 6) of revised manuscript.

Position in the respiratory cycle should be defined for all figures. Inspiration and Expiration are spans of time and the plots are of values at discrete time points.

Response: The positions in the respiratory for inhalation and exhalation are now defined in figure 4 ~ 9 captions (line 198, 202, 218, 221, 236 and 239) in the revised manuscript. This study focused on comparison between 50-year-old and 80-year-old, and two discrete time points (limited) were selected for comparison on several parameters.   

The points in Figure 10 indicate that simulations were performed for ages 60 and 70. What were the model parameters for these runs? Epsilon must be defined.

Response: For 50-, 60-, 70-, and 80-year-old, geometrical parameter (diameter and centerline length) decreased by 10% in a decade, and material properties increased by approximately 7% in a decade, respectively. These parameters were referred from the previous study reported by Kim et al.[12]. This is added in line 245 ~ 248 (page 11) of the revised manuscript. Epsilon is equivalent strain, and it allows any arbitrary three- dimensional strain state to be represented as a single positive strain value (scalar). This is now defined in line 243 ~ 248 (page 11) of revised manuscript.    

 “ε is equivalent strain and it allows any arbitrary 3-dimensional strain state to be represented as a single positive value (scalar)”.

Fig 10: ‘Material Properties Change Ratio’ and ‘Tissue Thickness Change Ratio’ must be defined. Please justify the choice of regression equation; if there were just 2 data points (age 50 and 80) as indicated in the methods section why not just fit a straight line?

Response: Material properties change ratio and tissue thickness change ratio are defined in line 248 ~ 251 (page 11) of revised manuscript. We used 4 data points (50-, 60-, 70-, and 80-year-old) for the regression equation, and it was interpolated using Akima interpolation method. This is added in line 243 ~ 248 (page 11) of revised manuscript.

Fig 11: I have a number of concerns with this figure. 1) it should be made clear what direction the changes are. I can interpret from the PV loop that we are looking at the change from 50 to 80 year old models (e.g. compliance is greater in the 80 YO model). 2) it does not seem like a fair comparison to work of breathing at two different tidal volumes. There will be more work at a greater tidal volume even in the same lung model. In order to rectify this the work of breathing should be provided as work per unit volume (as is the standard practice). Also, the effort to overcome elastic recoil (not just resistive forces or PV loop area) should be considered since the effort of inspiration is not recovered during passive expiration. The formula used for work of breathing and the calculated values should be provided. 3) there were obviously be more strain in the 80 YO model because the lung volume is twice as high, I would prefer to compare tissue strains at the same lung volumes. 4) which tissue strain (G20 or G23) is shown?

Response: Figure 11 is now changed in revised manuscript. 1) change of direction (from 50-year-old to 80-year-old) is now clearly indicated in figure 11 (between line 274 and 275 – page 12). 2) work of breathing is now defined as work (P-V loop area) per unit tidal volume in line 267. The formula used for work of breathing is provided in figure 11. 3) The maximum strain comparison between 80 and 50 yr old is shown in Fig. 11 and the strain in the 80-year-old model is lower due to stiff tissue properties. 4) tissue strain is shown for both G20 and G23 in figure 11. See the revised figure 11 (page 12) in the revised manuscript.

Line 245: The discussion of strain is confusing since we don’t know what lung volume we are discussing. Also, Fig 10 shows strain decreasing with age and Fig 11 shows strain increasing with age.

Response: As shown in revised Fig. 11, the strain decreases with age and this result is consistent with the trend shown in Fig. 10. Please see revised Fig. 11 and this is added in line 284 ~ 285 (page 12) of the revised manuscript.

Line 246: Does shear stress increase or decrease with age?

Response: Shear stress decreases with age. This is added in line 288-289 (page 12) of the revised manuscript.

Line 252: I do not see a reference [66] in the bibliography

Response: That reference [66] was not applied in Endnotes format. Now correct reference [46] is placed in line 297 (page 12) of the revised manuscript instead of [66].

Line 255-257: Please discuss the ‘insights into tissue strains and the impact on tissue/cell function’ that your model may provide. The / is not proper grammar.

Response: ‘insight into tissue strains and the impact on tissue cell function’ is discussed in line 300 ~ 302 (page 13) of the revised manuscript.

 ‘/’ is also now deleted in the revised manuscript.

Line 259: I don’t understand what ‘lower bifurcations were reduced by 10%, and enlarged by 40% in the alveolar sac for the aged’ means.

Response: the sentence is now restated in line 303 ~ 306 (page 13) of the revised manuscript.

Line 260: 23% change in what? volume delivered at a given P? Change in what direction?

Response: 23% increased for 80-year-old in slope of P-V curve (compliance). Volume delivered at a given pressure. This is restated in line 304 ~ 306 (page 13) of the revised manuscript.  

Line 261: does ‘lower bifurcations’ mean ‘distal airways’

Response: Yes, it does. The lower bifurcation means distal airways.

Line 263: Unless it’s shown that tidal volumes increase in aging I disagree with the statement about the work of breathing. Of course it takes more work to take a larger breath. The influence of the chest wall (which increases in stiffness with aging) on the work of breathing should also be mentioned.

Response: The influence of the chest wall stiffness on the work of breathing is now added in line 309 ~ 311 (page 13) of the revised manuscript.

Line 265: 40% change in what direction (increase or decrease with age)? At what lung volume?

Response: This line is removed due to unclarity in revised manuscript.

Line 265: observed should be predicted

Response: The term ‘observed’ is replaced with ‘predicted’ in line 310 (page 13) of the revised manuscript.

Line 266: Tissue stiffness seems like a prescribed parameter. Do the authors mean airway/alveolar stiffness (the stiffness of all types of tissue taken together)?

Response: Tissue stiffness means airway wall stiffness.

Line 266: Reference 48 seems to be in conflict with the statements in the introduction about tissue stiffness decreasing with age.

Response: Reference 48 in line 312 (page 13) is now changed into 12.

Line 278: What mechanical characteristics of the alveolar tissue?

Response: mechanical characteristics is strains in this study. Mechanical characteristics is replaced with strain in line 326 (page 13) of the revised manuscript.

Line 280: Since there were only two model runs how is it possible to have a 10-40% change in compliance? Fig. 11 seems to indicate that the compliance change is a single scalar value.

Response: Line 280 is now restated in lines 325-329 (page 13) of revised manuscript.

Line 281: I don’t understand what ‘This finding indicates the effect of strain change’ means

Response: Line 281 is restated in line 328 ~ 329 (page 13) of the revised manuscript.

Line 293: I would argue that the strain distributions were substantially impacted by the change in model parameters and not ‘the aging model was significantly impacted by the strain distributions’.

Response: Line 293 is now restated in line 342 (page 13) of the revised manuscript.

Line 294: the conclusion that ‘These strains can greatly damage the tissue leading to shearing and swelling’ is not supported by the data. First, it is not shown that strain leads to swelling. Second, it seems that shearing would be the deformation of the tissue by shear stress which the authors argue earlier will lead to improved barrier function. Finally, does the model show increased or decreased strain in the 80 year old?  

Response: We agrees reviewer’s comment, and line 294 is now deleted in revised manuscript. Finally the model shows decreased strain in the 80-year-old.

Line 297: Fig 10 shows decreased strain and Fig 11 seems to show increased strain with age. Which is it? Assuming that strain goes down, please discuss this finding in context of the increase sensitivity of the aged lung to VILI. 

Response: Strain in Fig 10 and Fig 11 (revised) indicates strain are decreasing. Strain of 80-year-old is lower than 50-year-old. Sensitivity of the aged lung to VILI is now discussed in line 323 ~ 329 (page 13) of revised manuscript.

Round 2

Reviewer 2 Report

The paper looks quite nice! There's one sentence in the abstract that a was still a bit confusing:

L25-L27 – This sentence is quite confusing. There is a 40% decrease in alveolar strain from 50 to 80 years old at inspiration and, at that point, alveolar pressures are 1.8 cmH2O lower in the 50 year old model? Likewise, there was a 27% decrease in bronchiole strain in the 80 year old compared to the 50 year old. The bronchial pressure was 10 cmH2O greater in the 80 year old model?

and a couple of really minor comments:

L46 – doesn’t lung injury cause a stiffening of the lung (decrease in compliance?)

L131 – zero velocity or zero pressure for the boundary condition?

Author Response

Thanks to the reviewer, and appreciate his/her thorough review of our manuscript.

The response is follows:

L25-L27 – Please see the rewritten sentence. The pressures information is reflected in lines L192-L194 rather than in the abstract to make it clear. Please see the revised manuscript.

L46 – This line is restated as “Ventilator-associated lung injury is severer due to an increase in lung compliance [3].

L131 – Corrected, it should be zero pressure for the boundary condition. See Lines 131-132 of the revised manuscript.